

# Using machine learning to disentangle LHC signatures of Dark Matter candidates

**Charanjit Kaur Khosa[1★], Veronica Sanz[1,2,3†] and Michael Soughton[1‡]**

**1** Department of Physics and Astronomy, University of Sussex, Brighton BN1 9QH, UK
**2** Departament de Física Teòrica and IFIC, Universitat de València-CSIC, E-46100, Burjassot, Spain
**3** Alan Turing Institute, British Library, 96 Euston Road, London NW1 2DB, UK

★ Charanjit.Kaur@sussex.ac.uk, † V.Sanz@sussex.ac.uk, ‡ M.Soughton@sussex.ac.uk

## Abstract

We study the prospects of characterising Dark Matter at colliders using Machine Learning (ML) techniques. We focus on the monojet and missing transverse energy (MET) channel and propose a set of benchmark models for the study: a typical WIMP Dark Matter candidate in the form of a SUSY neutralino, a pseudo-Goldstone impostor in the shape of an Axion-Like Particle, and a light Dark Matter impostor whose interactions are mediated by a heavy particle. All these benchmarks are tensioned against each other, and against the main SM background (Z+jets). Our analysis uses both the leading-order kinematic features as well as the information of an additional hard jet. We explore different representations of the data, from a simple event data sample with values of kinematic variables fed into a Logistic Regression algorithm or a Fully Connected Neural Network, to a transformation of the data into images related to probability distributions, fed to Deep and Convolutional Neural Networks. We also study the robustness of our method against including detector effects, dropping kinematic variables, or changing the number of events per image. In the case of signals with more combinatorial possibilities (events with more than one hard jet), the most crucial data features are selected by performing a Principal Component Analysis. We compare the performance of all these methods, and find that using the 2D images of the combined information of multiple events significantly improves the discrimination performance.

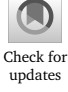

# 1 Introduction

After the Higgs boson discovery [1,2] at the Large Hadron Collider (LHC), some of the discovery potential focus has shifted towards Dark Matter (DM) searches. The discovery of DM and its characterisation would have profound consequences in Particle Physics, Cosmology and Astrophysics and the LHC could be the key to it. In spite of having experimental evidence of the presence of DM, we do not know what is its true nature, its mass scale, spin and interactions; or even if DM is a particle or a whole sector of new particles and interactions as in the SM.

The unknown properties of DM open the possibility of many different types of DM candidates being consistent with the DM relic density determination from the Cosmic Microwave Background (CMB), as well as other astrophysical constraints. To further explore DM, searches are being conducted in three main directions: underground experiments aiming to directly detect the interaction of DM with nuclei (*direct detection*) [3], astrophysical observatories searching for an excess of light or charged particles in the sky (*indirect detection*) [4], and collider searches for imprints of DM in collisions of protons and leptons (*collider searches*) [5].

Among the hypothesized DM candidates, the category of Weakly-Interacting Massive Particles (WIMPs) enjoy a privileged position, as a WIMP DM could in turn link to other issues plaguing the Standard Model (SM) of Particle Physics. Indeed, the WIMP *paradigm* is realised in many extensions of the SM, such as Supersymmetry (SUSY), where the WIMP is typically a new stable Majorana fermion with electroweak couplings and mass linked to the breaking of SUSY.

Typical DM collider searches are based on the idea that a stable and neutral particle, if produced at colliders, would leave the detector without resistance. Hence, the collider strategy is to search for traces of the DM presence via the associated production of other particles, namely the identification of singular objects within the detector (*mono searches*), where a

single object could be a jet, W or Z boson, top quark, photon or $t\bar{t}$ pair. The motivation for using these channels is that DM candidates (which cannot be directly detected) could be exposed through a momentum-mismatch in the final state, where the detected objects appear to recoil against nothing [6]. Note that collider stability is the most common assumption for DM searches but that a variety of DM scenarios could also be indirectly probed at the LHC via displaced track and vertices signatures using the long lived particle searches, see e.g. ref. [7] and references therein.

In this paper we focus on collider-stable particles, i.e. particles which do not interact with the detector and do not decay before leaving it, hence manifesting as missing momentum and mimicking DM particles. Note that a collider-stable particle could be completely stable (as is thought to be the case for DM particles) or be unstable but with a lifetime long enough such that it is not likely to decay before it leaves the detector. In our search for New Physics at the LHC, we must then attempt to remove this degeneracy. This task is even more challenging than analysing the signal and background for DM searches because here we are concerned with two or more (unknown) physics models.

The challenge of disentangling different DM candidates would likely benefit from the implementation of new and more sophisticated techniques, beyond the conventional search strategies. This difficult task calls for the use of Machine Learning (ML), which is emerging as a powerful tool for New Physics searches at LHC (see e.g. the recent review by Radovic *et al.* [8] and references therein).

More conventional ML methods like boosted decision trees (BDTs) have already been incorporated into many data analysis packages which made a significant impact in the analysis. Several recent studies have shown exciting applications of the ML methods for various tasks, like constraining Wilson coefficients of higher-dimensional operators in the EFT framework [9–11], top-tagging [12,13], cosmological phase transitions [14], parameter exclusion in SUSY models [15], quark-gluon tagging [16] etc. ML techniques have also been used recently for non-collider DM searches using substructure probes [17, 18], for cosmological DM [19] and in direct detection experiments [20].

In this work, we will apply ML techniques to explore the possibility of disentangling different scenarios for DM and impostors. Initially, we will focus on supervised methods and leave for the future a more model-independent unsupervised or semi-supervised explorations. We will consider the canonical search for DM: events with one jet recoiling against missing transverse energy (*monojet searches* [21, 22]), and use Machine Learning techniques in DM signal characterization.

The main focus of this paper is to explore the potential of ML techniques for the characterization of the discovered DM candidate [1]. Therefore our task is to explore the features of the signal and to be able to distinguish one signal from another. We will compare the features of DM signals from different Beyond the Standard Model (BSM) models. Specifically, we will use as benchmarks of comparison three types of models: a heavy WIMP dark matter from SUSY [23], Axion-Like Particles (ALPs) [24,25] and a simplified DM model with a spin-1 mediator. These will provide enough variety of characteristics to analyse differences and degeneracies among models.

Note we will base our analysis solely on differential information, not on overall cross-sections. The reason to restrict the analysis on kinematics alone is due to the freedom one has on the values of couplings in each model which impact the production cross section and the branching ratios to specific final states. For example, the production cross section of SUSY WIMPs depends strongly on its nature, e.g. Bino-like or Higgsino-like would lead to different values on the total number of events, yet differential kinematics (divided by the total number

---

[1]However in the appendix we also perform the classification task for the SM background versus signal benchmarks.

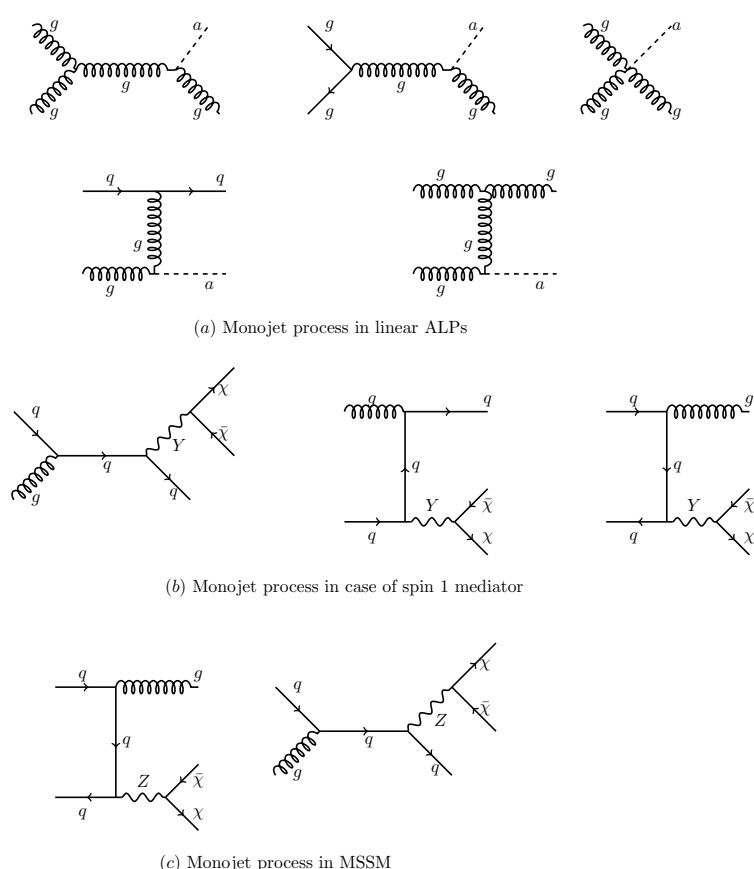

(a) Monojet process in linear ALPs

(b) Monojet process in case of spin 1 mediator

(c) Monojet process in MSSM

Figure 1: Feynman diagrams for monojet process in the linear ALP (top), EFT framework with a massive spin-1 mediator (middle) and SUSY WIMP DM cases (bottom).

of events) would not sizeably change. By restricting the analysis to differential information, we can draw more robust conclusions about the ability to distinguish different scenarios. Note though that discovery potential does depend on both discrimination power and production cross section, and this work focuses on the aspects of the analysis which can then be translated into different benchmarks.

The paper is organised as follows. In the next section 2 we describe the models which we use for benchmarking DM scenarios. In Sec. 3, we discuss kinematic features in the monojet signal, both at leading-order in the QCD expansion (LO) and next-to-leading-order (NLO) and show differences between the SUSY benchmark and the SM background. In section 4 we describe the ML methods and address the WIMP DM characterization using ML methods considered. In the last section, we discuss our findings and conclude.

## 2 Description of Benchmark Models

In this section we describe our choices of benchmark models. These models span a large enough range of kinematic features to compare with SM processes, as well as to see the

strength and limitations of the task of disentangling different DM scenarios.

Table 1: Summary of benchmark models.

| Model | Mass | Type of coupling |
|-------|------|------------------|
| SUSY1 | $m_{\tilde{\chi}^0} = 100$ GeV | Bino-like |
| SUSY2 | $m_{\tilde{\chi}^0} = 200$ GeV | Bino-like |
| SUSY3 | $m_{\tilde{\chi}^0} = 300$ GeV | Bino-like |
| ALP | negligible | gluon-ALP |
| EFT | negligible | 4-fermion |

We first consider a set of three benchmarks in the WIMP DM scenario based on Bino-like SUSY neutralinos and with masses in the range 100 to 300 GeV, see Table 1. We did not consider heavier WIMPs as they would likely be very hard to find at the LHC in monojet events. The cross section of production in monojet final states decreases quickly with the WIMP mass [26].

To contrast against the WIMP we consider two alternative cases. The first is an Axion-Like Particle (or ALP) which is a paradigm for signatures from pseudo-Goldstone bosons. They can be light and collider-stable due to the derivative (suppressed) nature of their couplings. ALPs themselves could be a DM particle or can be a DM mediator (see e.g. [27]). These exotic particles are constrained by Astrophysics as well as colliders in a complementary fashion [24, 28]. ALPs are also searched in axion experiments, which are designed to target their couplings with the photons.

In this work, we do not restrict the ALP to be a DM candidate, as it could decay after being produced, just not inside the detector [2]. It escapes detection as it has no charge under $SU(3)_c \times U(1)_Y$, hence its signatures are the same as those of DM in mono-searches. For the monojet channel, the ALP relevant couplings are to gluons, as couplings to quarks are mass-suppressed:

$$\mathcal{L}_a \supset -\frac{g_{agg}}{2} a \, \text{Tr}\left[ G_{\mu\nu} \tilde{G}^{\mu\nu} \right]. \tag{1}$$

Note that the ALP-gluon coupling has the following bound [24, 25]

$$g_{agg} \lesssim 1.1 \cdot 10^{-5} \, \text{GeV}^{-1} \quad (90\% \text{ C.L.}) \quad , \text{for} \quad m_a \lesssim 60 \, \text{MeV}. \tag{2}$$

Note that coupling of ALPs with photons or massive particles would lead to mono-photon, -W, -Z, -top and -Higgs signatures.

We consider a second alternative scenario to WIMP DM based on light DM produced from the decay of the heavy mediator. We label this EFT DM, as the effective interaction between the SM and DM is via a four-fermion higher-order operator. The simplified model Lagrangian which describes the interaction of DM ($\chi$) with the vector mediator ($Y$) is given by

$$\mathcal{L}_Y = \bar{\chi} \gamma_\mu g_\chi^V \chi Y^\mu. \tag{3}$$

and similarly for the interaction between the mediator and quarks. Note that the kinematic distribution of events is not very sensitive to the dark matter mass (see Fig. A1 in the appendix) in the limit of $m_Y \gg m_\chi$.

Within these models, monojet signatures would result from the processes shown in Fig. 1: a pair of DM particles (or a single ALP) produced in association with one initial state radiation (ISR) gluon or quark.

---

[2]If the ALP decays inside the detector, its detection would still be difficult due to its lightness. Nevertheless, one can still search for its effects on tails of distributions [29].



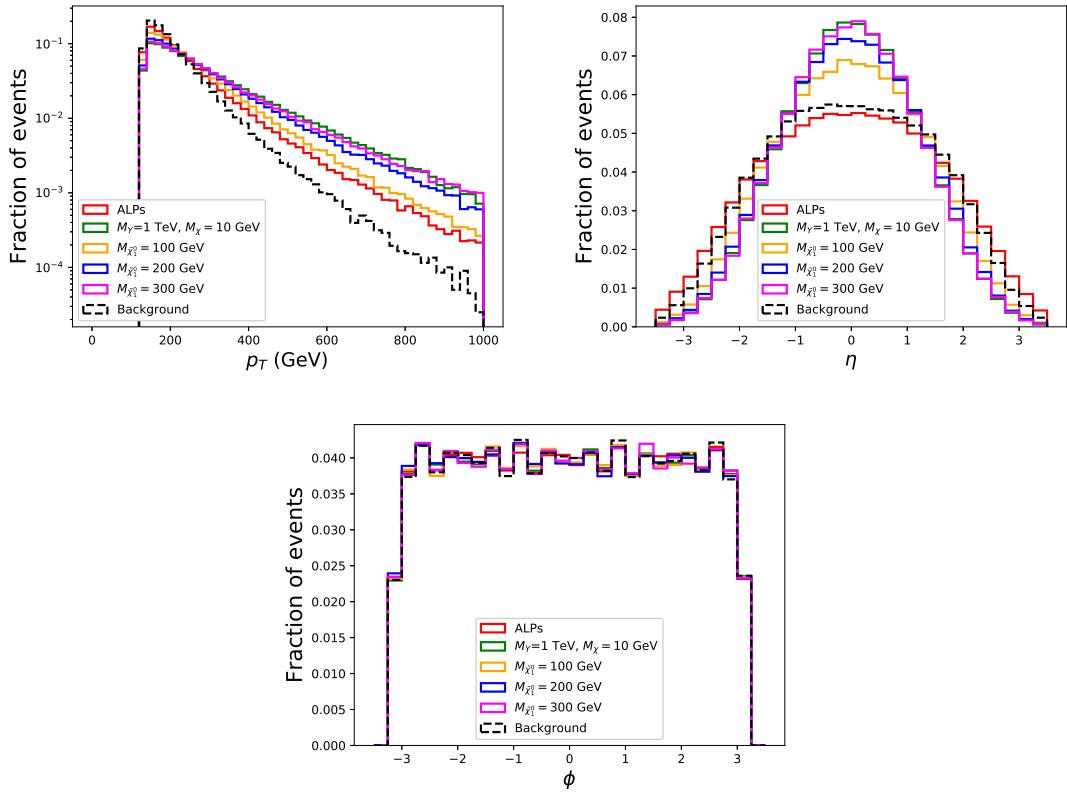

Figure 2: Monojet detector-level 1D histograms for WIMP, EFT and ALP scenarios, as well as the SM background.

Finally, we consider the dominant SM background given by $Z$+jets, where the $Z$ boson decays to neutrinos.

# 3  Kinematic Distributions

After presenting our benchmark scenarios for New Physics, in this section we describe the kinematic features which the Machine Learning algorithms will be able to optimize over. The aim of this section is to explain some of the features that the ML results will show.

The input for the ML studies is a list of events with their kinematic features simulated using a particular benchmark model. We perform both parton- and detector-level simulations to generate the data samples. More details related to event generation can be found in Appendix A.

The kinematic features of those events are generically multi-dimensional, but one can project into 1D and 2D kinematic space. For example, in Fig. 2 we show a set of 1D distributions for the monojet process [3]. Note that the jet transverse momentum in SUSY1 (lightest SUSY WIMP) and ALP is very similar, and also close to the SM background distribution, whereas the other SUSY scenarios and the EFT exhibit a harder spectrum and easier discrimination with respect to the SM. Additionally, the pseudorapidity $\eta$ distribution of the jet is very similar for the ALP and SM background cases. As the EFT case leads to harder $p_T$ spectrum and easier to differentiate from ALPs than the light SUSY case. We will see how the ML techniques will

---

[3]The overall behaviour is indeed maintained from parton-level to detector-level.

exhibit a similar trend. ALPs will be hard to separate from SUSY1, or light DM, whereas EFT and SUSY3 will have the most overlap, as both exhibit similar hard spectra.

At the level of 1D distributions, one cannot distinguish any preferred direction of the azimuthal angle $\phi$ distribution for both the New Physics signals and SM background processes.

Additional information can be obtained when moving from 1D to 2D distributions, as we will show in section 4.3 (where we show the event distribution in the 2D plane of $p_T^j$ vs $\eta$) and is discussed (in the Bayesian context) in Ref. [30]. Different models are compared to each other in this plane. The characteristics of these rough features, broadening in pseudorapidity and $p_T$ reach, and what conclusions one can draw from them, depend on level of accuracy of our simulation. To explore robustness against showering and detector effects, we promoted the simulation to the `Pythia` and `Delphes` level.

Finally, we discuss how one would use another source of information from these monojet topologies. Monojet events do often contain an additional hard jet. Splitting or additional ISR emission is not extremely rare, and with an additional object in the final state more information can be extracted of the DM nature. To account for the additional jet, we simulate detector-level events for the monojet (LO) and dijet (NLO) production. For all the cases, we consider data samples of 50K events with one additional jet of $p_T > 25$ GeV. 1D histograms for the constructed kinematic variable are shown in Fig. 3. An additional jet provides a new source of kinematic information, enhancing the discriminating power of our study. For example, notice that the distributions of $\Delta\phi_{j_1 j_2}$ ($j_1$: leading jet, $j_2$: sub-leading jet) and $\Delta\phi_{MET}^{j_2}$ are quite characteristic for the signals and background.

# 4 DM Characterization using Machine Learning

In this section we start our discussion on the use of supervised Machine Learning techniques for Dark Matter signal characterization, i.e. we focus on disentangling the different DM signals from each other rather than the discovery of any DM signal from the background. In other words, we are assuming that an excess of events has been identified at the LHC, and one is trying to unveil its true origin. We then ask the question: *Could there be contamination of the SUSY imposters in the vanilla SUSY WIMP events?* Assuming a similar size of signal events, the main question we shall endeavour to answer is then whether it is possible to disentangle an electroweak-scale WIMP DM signal from an ALP or an EFT signal with very light DM.

The input data we used for the analysis consists of three kinematic features of the monojet i.e. $p_T$, $\eta_j$ and $\phi_j$. We use this data both as an array input and also in the form of 2D histograms to build a ML algorithm.

In the following sections, we will compare the performance of different methods for parton-level and detector-level simulations. For this analysis, we will consider the kinematics of the leading jet ($p_T > 130$ GeV) for the monojet analysis and in addition to this also of the sub-leading jet for the dijet data sample. For the reasons mentioned in Sec. 2, we do not use the cross-section information, so a balanced dataset is considered for all the classes. Data samples are divided in 70% : 30% proportions for the training and test samples[4].

## 4.1 Logistic Regression

As a first step, we perform logistic regression for the monojet data without any data processing. We used SGDclassifier of SKLEARN python library with a log-loss function. For the parton level

---

[4]A portion of the data sample which will not be shown to the network while training. Rather, it will be used to assess the performance of the network as on the unseen data set.

Figure 3: Detector-level 1D histograms for the dijet process for WIMP, EFT and ALP scenarios, as well as the SM background.

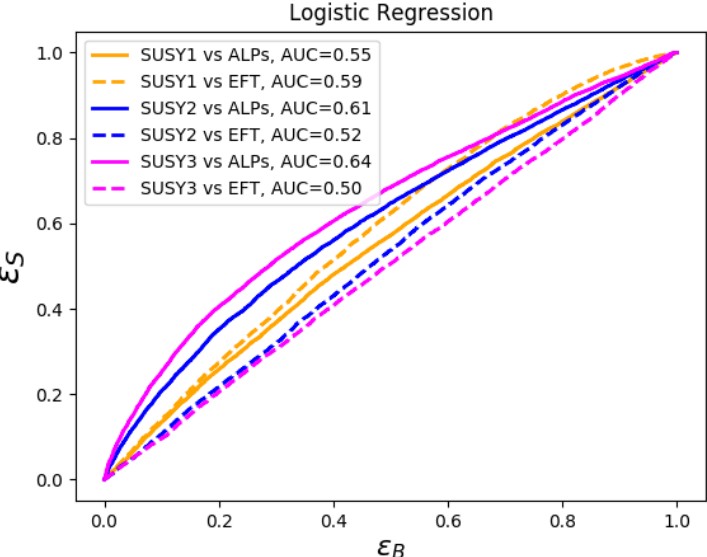

Figure 4: ROC curves using logistic regression for different WIMP scenarios versus ALP/EFT signals (monojet process).

events, Receiver operating characteristic (ROC) curves for SUSY signals vs ALP and EFT signals are shown in Fig 4. We consider the three benchmark values for the neutralino mass in the WIMP scenario described in Sec. 2. We can see that the AUC[5] value varies from 0.50 to 0.64 for the SUSY3 versus EFT and SUSY3 versus ALP, and the other four cases lie in-between. In other words, one can *easily* separate heavy WIMPs from the ALP monojet, however EFT monojet would not be efficiently classified. The regularization parameter $\lambda = 10^{-5}$ is used for all the cases.

## 4.2 Neural Networks-kinematic features

We investigate the classification accuracy using Deep Neural Networks (DNN) with the same input features i.e. $p_T$, $\eta$ and $\phi$ of the jet. We used five fully-connected hidden layers for the network as including more layers does not improve the performance.

For all the layers, except the final one, the number of neurons is equal to the number of data features that are considered. For the intermediate layers, a ReLU activation function is used, whereas we use a 'sigmoid' function for the output layer. We considered the binary cross-entropy loss function. The 'dropouts'[6] option is also activated, with 0.2 as the optimized choice. Finally, the 'adadelta' optimizer[7], and batch-size and epochs are set to 500 and 300, respectively.

Before discussing the performance of DNN for the signal classification, we discuss some aspects of the analysis when using DNNs with the three kinematic inputs: the first aspect relates to the robustness of the analysis with changes in the level of simulation detail. We see from the similarity between figures A3 and A4 in Appendix C for the signal versus background scenario that the DNN only performs slightly worse when using the more noisy detector-level data than the pure parton-level data. We also compared the performance of DNNs with just two variables (by excluding the $\phi$ variable) and found that there is no degradation in the classification accuracy. This is expected, as the three input variables are redundant once energy-momentum

---

[5]AUC (area under the ROC curve) is a measure of the algorithm's performance.
[6]It regularizes the network by deactivating the fraction of neurons.
[7]It is used to update NN weights for loss function minimization.

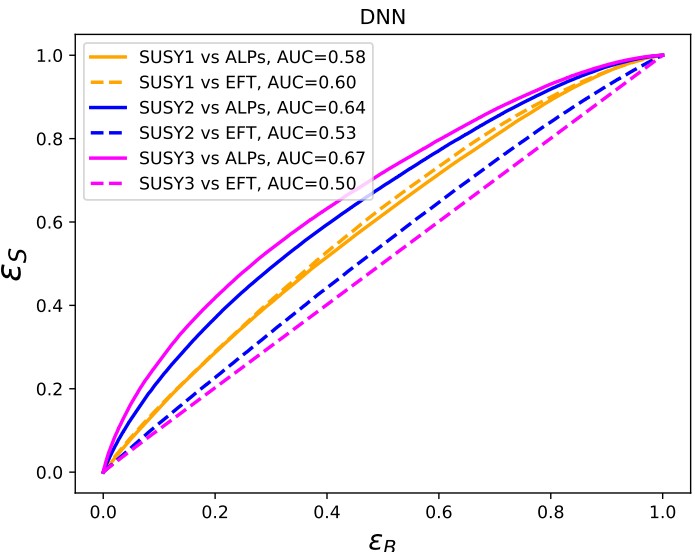

Figure 5: ROC curves using Neural Network for different WIMP scenarios versus ALP/EFT signals using the raw kinematic features (monojet process).

conservation within the event is taken into account[8]

We use DNNs with kinematic features as input for the classification of different signals. As mentioned earlier, classification accuracy is not very different for the detector level simulated data, hence we consider parton-level data in this section. In Fig. 5, we show the ROC curves using the Neural Network with kinematic features for the WIMP signal, where the other new physics signals are taken as competitors. The AUC values for WIMP SUSY3 versus EFT, and SUSY1 versus axion signal are 0.50 and 0.58 respectively (same as logistic regression case). Therefore the likelihood to misidentifying light WIMPs and axions is very high, and the same is true for heavy WIMP and EFT DM.

Another way to express this statement is shown in Fig. 6, where we plot the likelihood that a true WIMP signal is misidentified for an ALP or EFT DM scenario. The likelihood is computed as the false positive rate for the optimal point on the ROC curve. Since we are not considering the weight of the cross-section for different signals, we find the optimal point by minimizing the Euclidean distance from the (1=TPR[9],0=FPR) value.

## 4.3 DNN with 2D histograms

As mentioned earlier, the information in the monojet event is saturated by choosing two variables, $p_T^j$ and $\eta^j$. Therefore, inspired by the use of Convolutional Neural Networks (CNNs) in the classification of images, we construct 'images' from 2D histograms using $p_T$ and $\eta$ of the jet.

One has choices on how to group the events into images. The simulated dataset contains $N_{\text{Tot}}$ total number of events, and is divided into $N_{\text{Images}}$ number of images, such that each image contains $r = N_{\text{Tot}}/N_{\text{Images}}$ number of events.

---

[8]We also explore the benefit of considering dijet processes in addition to monojet for the signal versus background scenario. As mentioned previously, dijet searches have the potential to provide more information on the nature of DM particles. In this case we consider eight features: $p_T^{j_1}, p_T^{j_2}, \eta^{j_1}, \eta^{j_2}$, MET, $\Delta\phi_{j_1 j_2}, \Delta\phi_{MET}^{j_1}$, and $\Delta\phi_{MET}^{j_2}$. Using the DNN with the same architecture for this data sample we get indeed an enhancement in the classification accuracy as shown in Fig. A5 in Appendix C.

[9]TPR is fraction of signal events correctly identified by the algorithm, and FPR is the fraction of background events identified as signal events by the algorithm.

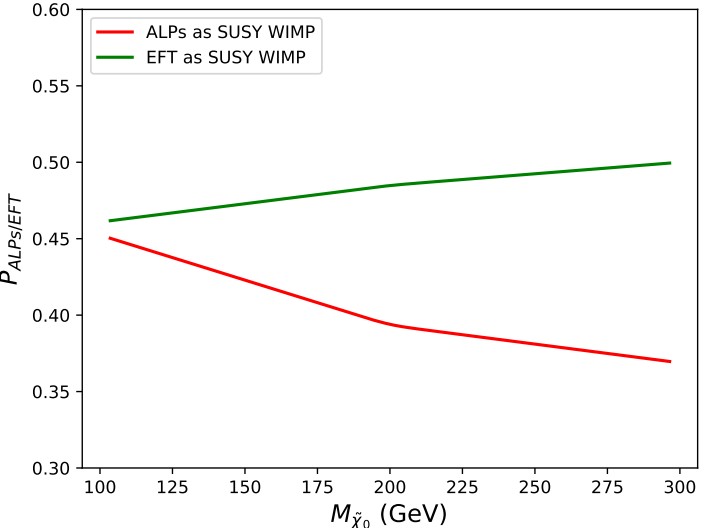

Figure 6: Probability of misidentifying ALPs or EFT DM scenarios with a true SUSY WIMP as a function of the neutralino mass.

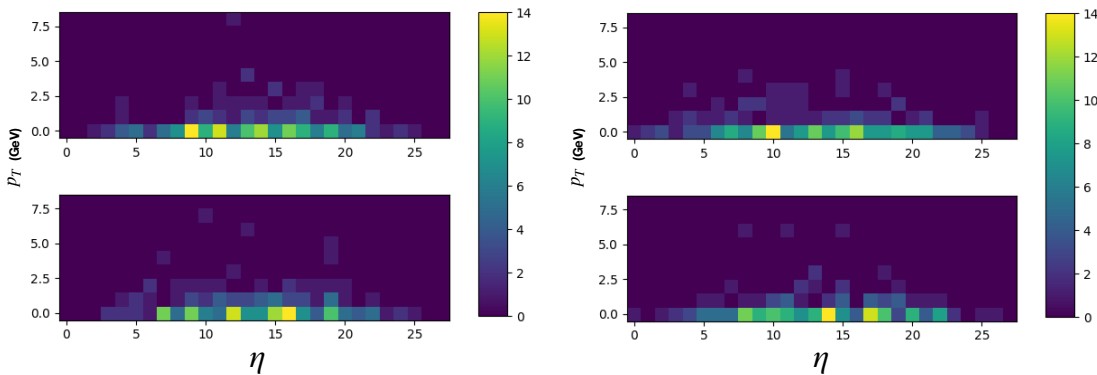

Figure 7: Illustrative 2D histograms for SUSY1 (left) and ALP case (right) averaging over 200 events.

Creating a number of 'images' to train a network is a powerful tool since each image itself is giving an approximation of the joint Probability Density Function (PDF) distribution for both $p_T$ and $\eta$. The degree to which each image approximates the true theoretical PDF depends on the number of events $r$ chosen to be in an image. For a fixed total number of events (determined by the LHC integrated luminosity) there will then be a trade-off between $r$ and $N_{\text{Images}}$ that will affect the accuracy of the model. We shall see that by viewing the input data in a different form, the algorithm is able to learn more information of the distributions. We will examine this method in both the monojet case and the dijet case.

### 4.3.1 Monojet

Before analysing the data with a CNN, we first attempt solving the problem with a DNN. This is achieved by decomposing the 2D histogram into a 1D array with values corresponding to the normalised number of events in each bin. Note that whilst this may seem like we are just reconstructing the original distributions, this is not the case since this data now contains

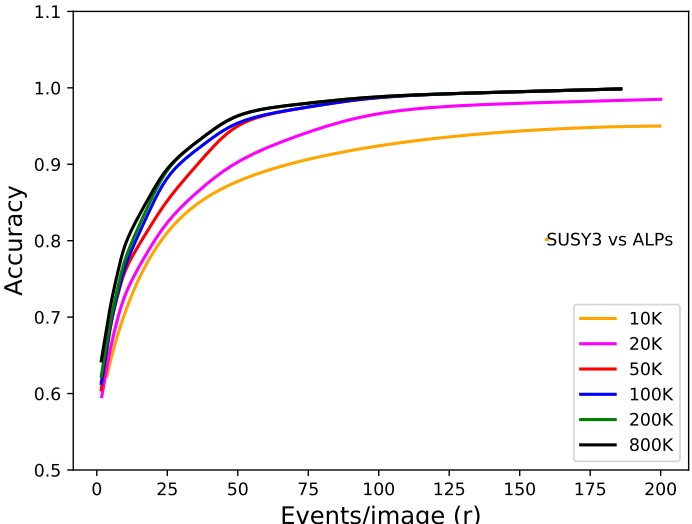

Figure 8: Accuracy versus number of events per image, with varying total number of events for SUSY3 versus ALP. We use fully-connected NNs with two hidden layers for different data sample size. The monojet data set is considered for this plot.

correlations from individual distributions. A few illustrative pictures of these plots are shown in Fig. 7.

We consider $p_T$ and $\eta$ in range [130, 2000] GeV and [-4 to 4], respectively with 29 × 29 bins. We use the information of event density in this grid as an input for a DNN. The network is well optimised with two fully connected hidden layers, both consisting of twenty neurons with a ReLU activation function and a *softmax* activation function for the output layer. After investigation, we do not find over-training to be an issue, so we only include a small number of dropout neurons. The network is trained for 300 epochs with a batch size of 500. We evaluate the network's performance through accuracy (found from finding whether the predicted result (ranging from 0 to 1) round to 0 or to 1). This is done within the training dataset whilst training, then with the validation dataset whilst evaluating how the network is training, and finally with the test dataset.

We find that averaging over more number of events per image improves the classification accuracy. As an illustration of this effect, in Fig. 8, we show the accuracy plot for WIMP SUSY3 versus ALPs data and how it depends on the number of events injected in one picture for the fixed data sample. The different colored curves correspond to varying the total data sample. For a fixed data sample, the number of images a DNN is trained and tested is reduced when we increase the number of events per image. As we know that with more events per image the input data approaches closer to the *theoretical* PDF for the event distributions and this is clearly reflected in the increase in accuracy with number of events per image. Nevertheless, at about 50 events per image, the accuracy plateaus and no much more gain is obtained by bunching more events per image.

We also studied the dependence of the accuracy with the size of the data sample. Once we have enough data to train the DNN, adding more events does not improve accuracy, whereas the accuracy decreases for smaller values of the ratio $r$. Clearly, for larger values of $r$, the increase of sample size from 10K to 800K is more important. The only limitation of the sample size is Monte Carlo generation, and one can observe no substantial differences once the sample size is above 100K.

We find that for $r = 1$ we obtain the same result as for the DNN using directly the kinematic

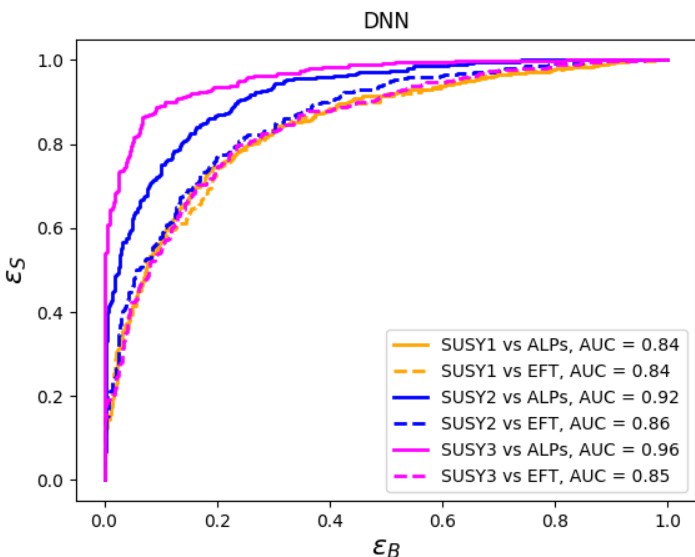

Figure 9: ROC curves (DNN with 2D histograms, $r = 20$) for SUSY WIMP benchmarks versus other signals, for the monojet data set (parton-level) with 50K events are used for each signal.

features. However for $r > 1$ we see a clear improvement in performance, due to feeding the network a different format of the data which allows the algorithm to learn more from the type of events present in each BSM model. This bunching of events $r \neq 1$ could be seen as a way to connect event-by-event analyses with PDF distributions of the data in the 2D plane.

In Fig. 8 central values of accuracy are plotted for different total number of events. With total number of 100K data sample, the DNN performance is very stable, but it starts becoming unstable for the 50K data sample. We also note that using less than 20k total events leads to large errors in accuracy. We do not show the accuracy for values of $r > 200$ since we are keeping the total number of events fixed, there are not enough images to produce reliable results. ROC curves for all signal-to-signal combinations using DNN with 2D histograms are shown in Fig. 9 for the monojet data sample.

As mentioned earlier, the analysis has been conducted with the assumption that the number of different type of signal events are the same and that we can fully decompose events into purely signal and background. This has allowed us to investigate disentangling the various DM model presented, however it is not a realistic scenario. We therefore also explore how effectively the classifiers can disentangle DM models with varying levels of signal in Appendix D.

### 4.3.2 Dijet

We also consider the Next-to-Leading-Order (NLO) processes in which two parton-level jets in addition to the 1-jet process are produced. We also simulate detector-level for the dijet processes, and use both the leading and sub-leading jet. More details about how two jet events are selected can be found in the Appendix A.

Since for the dijet process we have eight different features, there are 28 possible variations of 2D histograms that we could produce. If this were to be fully taken into account, then it might prove fruitful to combine the results of those 28 ML algorithms together in some manner. Alternatively, one could produce three (or higher)-dimensional histograms and deconvolve them for the DNN, or use a three (or higher)-dimensional CNN. However, we shall not take

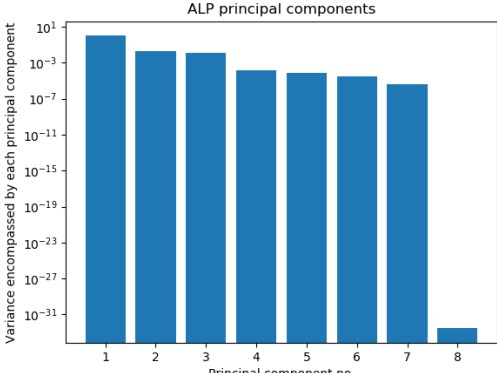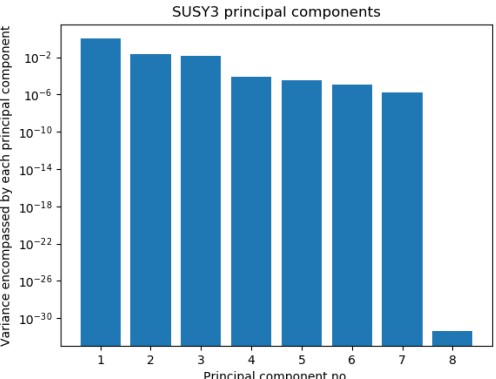

Figure 10: The explained variance ratio of the new principal component axes (with the number of principal component axes being 8, the same number of original features), highlighting the relative importance of the principal components in terms of capturing variance in the datasets.

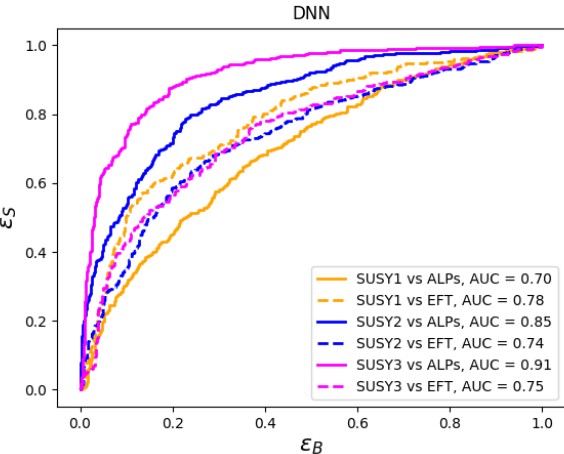

Figure 11: ROC curves (DNN with 2D histograms, $r = 20$) for SUSY WIMP versus signal for the dijet detector-level simulation. For these plots, 50K events are used for each process.

this approach here to avoid over-complicating the analysis and obscuring the physical insight which is our primary aim. Instead, we perform a Principal Component Analysis (PCA) to the dijet data in order to determine which features are more relevant.

Figure 10 (and Fig. A2 in the Appendix) shows that whilst all but one feature columns contribute to the variance of the dataset, a significant amount of the variance within the datasets is captured within the first three principal component axes. By finding the correlation between the original features and the new principal component axes, we can determine which features are most important in terms of holding information. We expect an increase in classification performance if we were to use the principal component axes as new feature columns, however we choose instead to use the original feature columns that correspond most strongly to the most important principal component axes, as we are primarily interested in establishing which features are most relevant for our 2D analysis as a proof-of-concept.

The correlations can be seen in Table 2 for ALP and the SUSY3 benchmark, and for other cases in Tables B1, B2, B3, and B4 of the Appendix B. One can see from the table that the three most important features are the same for all datasets, namely $p_T^{j_1}$, $\Delta\phi_{j_1 j_2}$, and $\Delta\phi_{\mathrm{MET}}^{j_1}$.

Table 2: PCA original feature to principal component correlations for the ALP and SUSY3 benchmark (rounded to 2 d.p.).

| | $p_T^{j_1}$ | $p_T^{j_2}$ | $\eta_{j_1}$ | $\eta_{j_2}$ | $\Delta\phi_{j_1 j_2}$ | MET | $\Delta\phi_{\mathrm{MET}}^{j_1}$ | $\Delta\phi_{\mathrm{MET}}^{j_2}$ |
|---|---|---|---|---|---|---|---|---|
| ALP PCA correlations | | | | | | | | |
| PC-1 | 0.66 | 0.36 | 0.01 | 0.00 | -0.01 | 0.67 | 0.01 | 0.01 |
| PC-2 | -0.01 | 0.00 | -0.02 | -0.02 | -0.38 | -0.01 | 0.76 | 0.52 |
| PC-3 | 0.01 | -0.01 | 0.05 | 0.04 | -0.76 | 0.00 | 0.06 | -0.64 |
| PC-4 | 0.00 | 0.00 | -0.70 | -0.71 | -0.04 | 0.00 | -0.02 | -0.06 |
| PC-5 | -0.29 | 0.93 | 0.05 | -0.04 | -0.01 | -0.22 | -0.01 | -0.01 |
| PC-6 | 0.02 | -0.06 | 0.71 | -0.71 | 0.00 | 0.01 | -0.00 | 0.00 |
| PC-7 | 0.70 | 0.05 | 0.00 | -0.00 | 0.00 | -0.71 | -0.00 | 0.00 |
| PC-8 | 0.00 | -0.00 | 0.00 | 0.00 | 0.52 | -0.00 | 0.64 | -0.56 |
| SUSY3, $M_{\tilde\chi_1^0} = 300$ GeV PCA correlations | | | | | | | | |
| PC-1 | 0.67 | 0.32 | 0.00 | 0.01 | 0.00 | 0.67 | -0.00 | -0.00 |
| PC-2 | -0.00 | -0.01 | 0.01 | -0.01 | 0.32 | -0.00 | -0.76 | -0.57 |
| PC-3 | 0.00 | -0.00 | -0.07 | -0.06 | -0.80 | 0.00 | 0.11 | -0.58 |
| PC-4 | -0.01 | 0.02 | 0.70 | 0.70 | -0.07 | -0.01 | 0.01 | -0.05 |
| PC-5 | -0.22 | 0.95 | -0.00 | -0.02 | 0.00 | -0.23 | -0.01 | -0.00 |
| PC-6 | -0.00 | 0.01 | -0.71 | 0.71 | 0.01 | -0.00 | -0.01 | -0.00 |
| PC-7 | 0.71 | -0.01 | -0.00 | 0.00 | -0.00 | -0.71 | 0.00 | 0.00 |
| PC-8 | 0.00 | 0.00 | 0.00 | -0.00 | 0.50 | 0.00 | 0.64 | -0.58 |

We do find an increase in classifier performance when using the $p_T^{j_1}$ and $\Delta\phi_{\mathrm{MET}}^{j_1}$ distributions over $p_T^{j_1}$ and $\eta_{j_1}$. The ROC curves for a DNN (with the same architecture as before) are shown in Fig. 11.

We note that using dijet data with the 2D histogram approach performs slightly worse than when using the monojet data showing that more information is contained within the $p_T^{j}$ and $\eta_j$ for a monojet process than the $p_T^{j_1}$ and $\Delta\phi_{\mathrm{MET}}^{j_1}$ of one jet in a dijet process and therefore it is still useful to include the other features of a dijet event, even if they do offer diminishing returns on performance after the second. Note that our results are not intended as a comprehensive evaluation but rather as evidence that it is possible and prudent to include dijet events for the 2D histogram method.

## 4.4 CNN with 2D histograms

After transforming our events into histograms/images (which are the approximations of PDFs) and processing them using a DNN, we move onto applying CNNs to them.[10]. We use a CNN with two convolutional layers, two max-pooling layers and one dense flatten hidden layer, with ReLu activation function for all the cases. ROC curves are shown in Figs. 12 and 13, for monojet and dijet processes respectively. As mentioned earlier, CNNs are able to retain the information of spatial correlations and usually perform better than DNNs for the image data. But in our case, because images are constructed from highly processed information (PDFs are already an abstract concept) we find no sizeable improvement respect to the DNN results.

---

[10]CNNs have been used in many HEP analyses, see e.g. [31, 32]

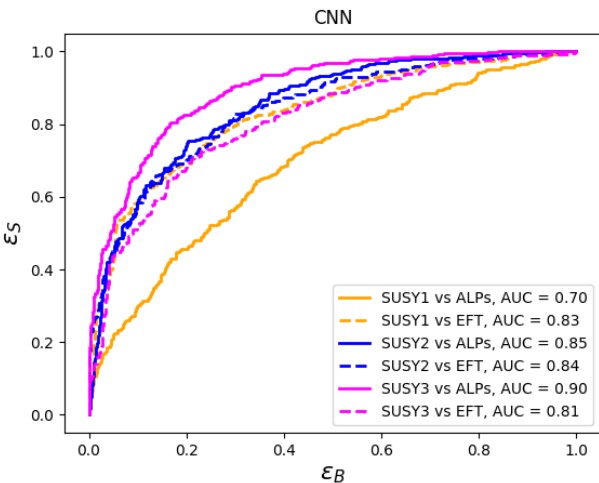

Figure 12: ROC curves (CNN with 2D histograms, $r = 20$) for SUSY WIMP benchmarks versus other signals, for the monojet case with 50K events are used for each signal.

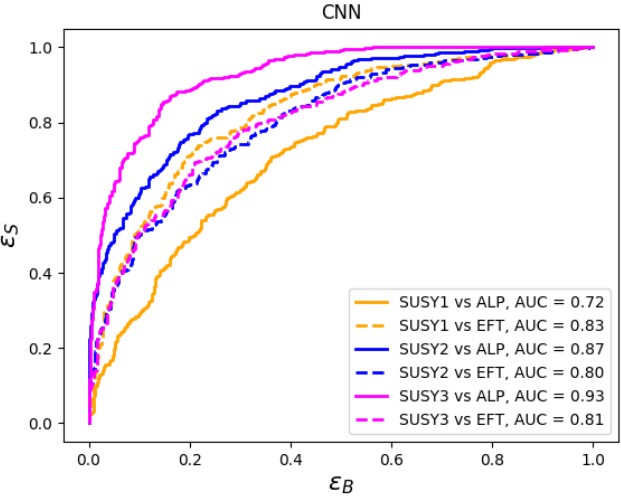

Figure 13: ROC curves (CNN with 2D histograms, $r = 20$) for signal versus signal for the dijet simulation. For these plots, 50K events are used for each process.

## 5 Conclusions and Outlook

In this paper, we explored supervised Machine Learning algorithms to disentangle different DM candidates, assuming an excess is observed in the channels involving a jet recoiling against missing transverse energy. We also considered the topology including an additional hard jet. For this analysis, we compared simulations with and without detector-level effect, and found that there is no significant degradation in the discrimination performance when we account for realistic detector-level effects.

We used different ML approaches to the classification problem: logistic regression, DNNs and CNNs. We found that for the kinematic variables NN performs slightly better than logistic regression.

We then created histograms with variable number of events, which can be seen as approxi-

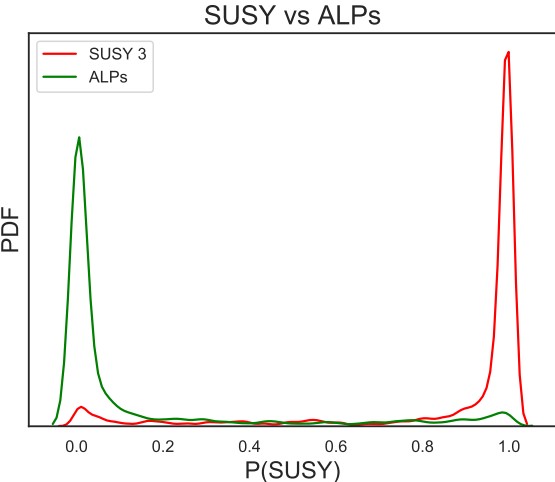

Figure 14: Classifier output P(SUSY) from the DNN run for monojet SUSY3 and ALP with $r = 20$.

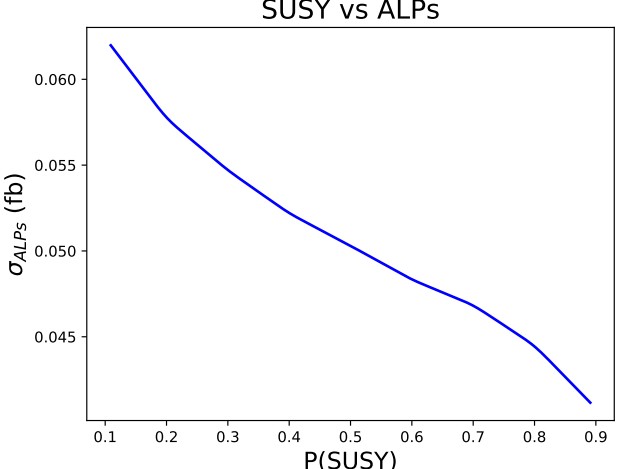

Figure 15: ALP cross-section bound for $\sigma_{SUSY} = 0.1$ fb and $S_{SUSY}/S_{ALP}=2$.

mations to the probability density of events in a kinematic 2D plane. We feed these histograms to a DNN and a CNN. The processing of these histograms offers a better classification accuracy than lists of events with values of kinematic observables. We show that this method can produce good results for the characterisation of different DM models. We found a trade-off between the accuracy and number of events averaged for one histogram ($r$). In a more realistic situation, one would expect a small sample of DM events, and therefore the question of how learning changes at low values of $r$ is key.

We also investigated processes with an additional hard jet to estimate the accuracy gain as compared to the monojet topology, as these topologies could contain more information despite a lower cross section. In this case, we performed a PCA analysis to decide the most important combination of features, and then constructed 2D histograms of those although we find that when using the 2D histogram method we obtain slightly worse performance when using dijet events with only two variables however we do demonstrate that it is possible to incorporate the additional information from dijet events into our methods.

One can translate the techniques proposed here into limits of impostor cross section by

incorporating the information of the cross-section of specific model benchmarks. An example on how one could translate this analysis into a limit-setting procedure can be understood as follows. In Fig. 14, we shows the classifier output of one of the algorithms we have trained, the DNN run for a choice of $r$. This algorithm output can be re-interpreted as an impostor cross-section limit, see Fig. 15. To produce this figure we assumed a fixed value of the WIMP cross section and plotted the impostor cross section which would be excluded using a similar analysis of Ref. [33].

Moreover, this analysis could be extended for other channels in collider dark matter searches, e.g. for the Vector Boson Fusion (VBF) topology where we have two additional forward jets. It would be interesting to compare the performance of proposed methods for the dijet and VBF case. A combined analysis of different channels may offer enhanced sensitivity which will be investigated in the near future.

Finally, the promising results of this analysis adds a further motivation to explore the possibility of using unsupervised techniques.

# 6 Acknowledgments

C.K.K. wishes to acknowledge support from the Royal Society-SERB Newton International Fellowship (NF171488). The work of V.S. is supported by the Science Technology and Facilities Council (STFC) under grant number ST/P000819/1. M.S. wishes to acknowledge the support by the Data Intensive Science Center in the South East Physics Network (DISCnet), an extension of the STFC, under grant number ST/P006760/1. We would also like to thank Alexander Belyaev for his comments on the levels of signal and background.

# A Analysis set-up

## A.1 Leading-order(LO) parton-level simulation

We generate parton-level events for monojet and missing energy signals and centre-of-mass energy $\sqrt{s} = 14$ TeV using MadGraph_aMC@NLO v2.6.3.2 [34]. For SUSY WIMP, we used MSSM-SLHA2 model (all the SUSY spectrum is set to be very heavy except the lightest neutralino). We use the Feynrules model file [35, 36] for the linear ALPs [37] and spin-one mediator case (DMsimp_s_spin1 [38] model) in the EFT framework.

Using these models, we generate the following processes

$$pp \rightarrow aj \qquad \text{ALPs} \tag{4}$$

$$pp \rightarrow \chi\bar{\chi}j \qquad \text{EFT, spin-1 mediator} \tag{5}$$

$$pp \rightarrow \tilde{\chi}_1^0 \tilde{\chi}_1^0 j \qquad \text{SUSY-WIMP}. \tag{6}$$

For SM monojet background, we consider the following dominant background:

$$pp \rightarrow Zj(Z \rightarrow \nu\bar{\nu}). \tag{7}$$

For all the processes 400K events are generated using a cut of $p_T > 130$ GeV for the jet $p_T$. The following kinematic variables are constructed;

$$p_T^j(\text{MET}), \quad \eta^j, \quad \phi^j.$$

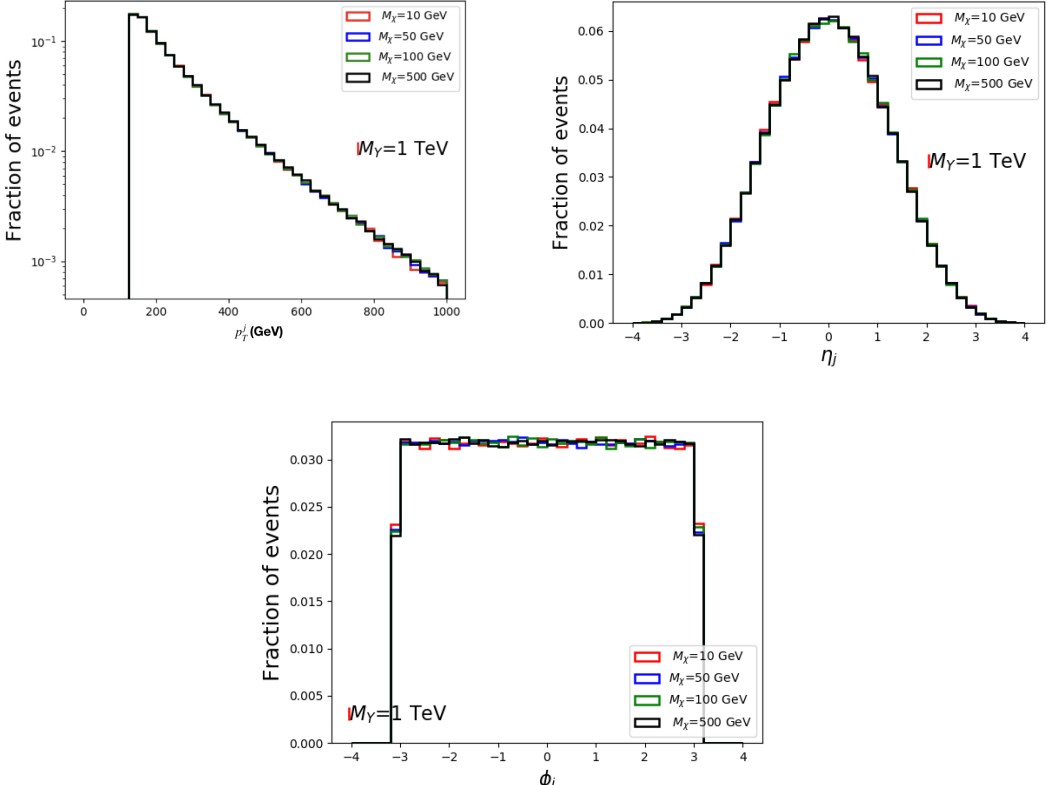

Figure A1: The $p_T$, $\eta$, and $\phi$ distributions for the monojet and MET process in case of EFT spin-1 mediator (at parton-level) for fixed mediator mass and various dark matter masses.

## A.2 Leading-order and Next-to-Leading order (NLO) detector-level simulation

We generate both monojet and dijet + MET processes. As earlier, hard processes are generated using Madgraph and PYTHIA [39] is used for hadronization and showering. Detector effects are incorporated using DELPHES [40] with the default ATLAS card[11]. A generation level cut of $p_T^j > 130$ GeV is used for leading jet $p_T$. For the dijet case, while analysing the root file, in addition to the leading jet $p_T$ cut we also demand $p_T^j > 25$ GeV for the sub-leading jet.

We have used 200K events for all the cases after root file analysis. To avoid the double-counting from showering, a jet merging scheme (MLM) with xqcut 20 GeV is applied. For the monojet, we consider the same three kinematic features as in the parton-level monojet events. For the dijet, we construct the following kinematic variables :

- $p_T^{j_1}$: transverse momentum of the leading jet.

- $p_T^{j_2}$ transverse momentum of the sub-leading jet.

- $\eta^{j_1}$: pseudo-rapidity of the leading jet.

- $\eta^{j_2}$: pseudo-rapidity of the sub-leading jet.

- MET: missing transverse momentum.

---

[11]As a check we compared our background simulation ($\sqrt{s} = 13$ TeV) with the latest ATLAS monojet paper [21] and found out that we get 15% more events than reported in the ATLAS paper.

- $\Delta\phi_{j_1 j_2}$: angular separation between the leading and sub-leading jet.

- $\Delta\phi_{\mathrm{MET}}^{j_1}$: angular separation between the MET and leading jet.

- $\Delta\phi_{\mathrm{MET}}^{j_2}$: angular separation between the MET and sub-leading jet.

For the dijet sample, we consider 50K events for all the processes.

# B    PCA correlations

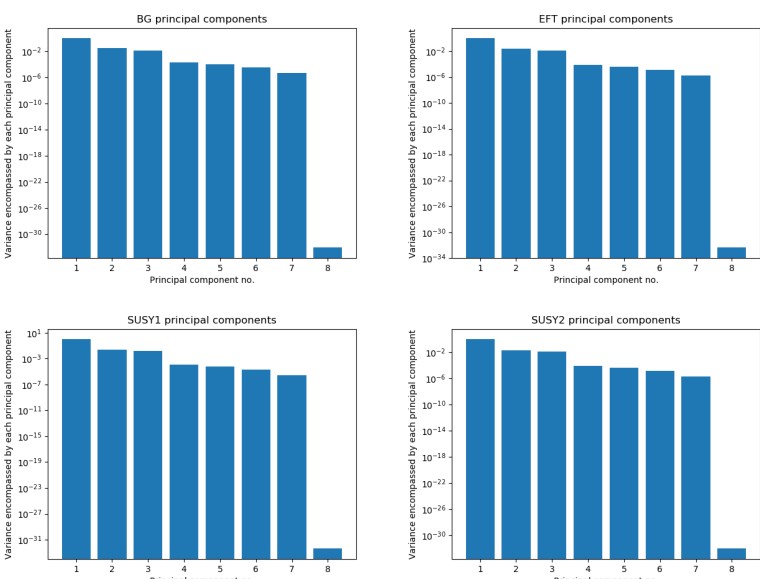

Figure A2: The variance ratio of the new principal component axes (with the number of principal component axes being 8, the same number of original features), highlighting the relative importance of the principal components in terms of capturing variance in the datasets.

Table B1: PCA original feature to principal component correlations for the SM background (rounded to 2 d.p.).

| Background PCA correlations | | | | | | | | |
|---|---|---|---|---|---|---|---|---|
| | $p_T^{j_1}$ | $p_T^{j_2}$ | $\eta_{j_1}$ | $\eta_{j_2}$ | $\Delta\phi_{j_1 j_2}$ | MET | $\Delta\phi_{\mathrm{MET}}^{j_1}$ | $\Delta\phi_{\mathrm{MET}}^{j_2}$ |
| PC-1 | 0.67 | 0.41 | -0.01 | -0.01 | -0.01 | 0.62 | 0.00 | -0.00 |
| PC-2 | 0.00 | 0.00 | 0.01 | 0.00 | 0.45 | -0.00 | -0.77 | -0.45 |
| PC-3 | -0.00 | -0.00 | 0.09 | 0.10 | -0.70 | -0.00 | 0.00 | -0.70 |
| PC-4 | -0.01 | 0.00 | -0.70 | -0.70 | -0.09 | -0.01 | -0.01 | -0.10 |
| PC-5 | -0.12 | 0.89 | -0.01 | 0.02 | -0.00 | -0.45 | 0.00 | 0.00 |
| PC-6 | -0.01 | 0.01 | 0.71 | -0.71 | -0.01 | -0.01 | 0.01 | -0.00 |
| PC-7 | 0.73 | -0.23 | 0.00 | 0.00 | 0.00 | -0.65 | 0.00 | 0.00 |
| PC-8 | 0.00 | 0.00 | -0.00 | -0.00 | 0.54 | -0.00 | 0.64 | -0.54 |

Table B2: PCA original features to principal component correlations for the EFT simplified framework (rounded to 2 d.p.).

| | $p_T^{j_1}$ | $p_T^{j_2}$ | $\eta_{j_1}$ | $\eta_{j_2}$ | $\Delta\phi_{j_1 j_2}$ | MET | $\Delta\phi_{\text{MET}}^{j_1}$ | $\Delta\phi_{\text{MET}}^{j_2}$ |
|---|---|---|---|---|---|---|---|---|
| | | | | EFT PCA correlations | | | | |
| PC-1 | 0.67 | 0.34 | -0.00 | 0.00 | 0.00 | 0.66 | 0.01 | 0.02 |
| PC-2 | -0.01 | -0.01 | -0.00 | 0.01 | -0.33 | -0.01 | 0.76 | 0.56 |
| PC-3 | 0.01 | -0.00 | 0.00 | 0.01 | -0.80 | 0.01 | 0.10 | -0.59 |
| PC-4 | -0.00 | 0.01 | 0.71 | 0.71 | 0.01 | -0.00 | -0.00 | 0.00 |
| PC-5 | -0.22 | 0.94 | -0.03 | 0.02 | -0.01 | -0.27 | 0.00 | -0.00 |
| PC-6 | 0.01 | -0.04 | -0.71 | 0.71 | 0.01 | 0.01 | -0.01 | 0.00 |
| PC-7 | -0.71 | 0.04 | 0.00 | -0.00 | -0.00 | 0.70 | -0.00 | -0.00 |
| PC-8 | -0.00 | 0.00 | -0.00 | 0.00 | 0.51 | 0.00 | 0.64 | -0.57 |

Table B3: PCA original features to principal component correlations for the SUSY1 case (rounded to 2 d.p.).

| | $p_T^{j_1}$ | $p_T^{j_2}$ | $\eta_{j_1}$ | $\eta_{j_2}$ | $\Delta\phi_{j_1 j_2}$ | MET | $\Delta\phi_{\text{MET}}^{j_1}$ | $\Delta\phi_{\text{MET}}^{j_2}$ |
|---|---|---|---|---|---|---|---|---|
| | | | SUSY1, $M_{\tilde\chi_1^0} = 100$ GeV PCA correlations | | | | | |
| PC-1 | 0.67 | 0.35 | 0.01 | 0.00 | -0.00 | 0.66 | 0.01 | 0.00 |
| PC-2 | 0.00 | 0.01 | -0.01 | -0.01 | 0.37 | 0.00 | -0.76 | -0.53 |
| PC-3 | 0.00 | -0.02 | -0.01 | -0.01 | 0.77 | 0.01 | -0.07 | 0.63 |
| PC-4 | -0.00 | -0.01 | 0.71 | 0.71 | 0.01 | -0.00 | -0.01 | -0.00 |
| PC-5 | -0.21 | 0.93 | -0.03 | 0.04 | 0.01 | -0.28 | 0.00 | 0.01 |
| PC-6 | -0.01 | 0.05 | 0.71 | -0.71 | 0.00 | -0.02 | -0.00 | -0.00 |
| PC-7 | 0.71 | -0.05 | -0.00 | -0.00 | 0.00 | -0.70 | -0.00 | 0.00 |
| PC-8 | 0.00 | 0.00 | 0.00 | 0.00 | -0.52 | -0.00 | -0.65 | 0.56 |

Table B4: PCA original features to principal component correlations for the SUSY2 case (rounded to 2 d.p.).

| | $p_T^{j_1}$ | $p_T^{j_2}$ | $\eta_{j_1}$ | $\eta_{j_2}$ | $\Delta\phi_{j_1 j_2}$ | MET | $\Delta\phi_{\text{MET}}^{j_1}$ | $\Delta\phi_{\text{MET}}^{j_2}$ |
|---|---|---|---|---|---|---|---|---|
| | | | SUSY2, $M_{\tilde\chi_1^0} = 200$ GeV PCA correlations | | | | | |
| PC-1 | 0.67 | 0.32 | 0.00 | -0.00 | -0.01 | 0.67 | 0.01 | 0.01 |
| PC-2 | 0.01 | 0.01 | 0.00 | -0.01 | 0.34 | 0.01 | -0.76 | -0.56 |
| PC-3 | -0.00 | 0.01 | 0.03 | 0.06 | 0.79 | -0.00 | -0.09 | 0.60 |
| PC-4 | 0.00 | -0.01 | 0.71 | 0.70 | -0.05 | 0.00 | 0.00 | -0.04 |
| PC-5 | -0.22 | 0.95 | 0.01 | 0.01 | -0.01 | -0.24 | 0.00 | -0.01 |
| PC-6 | 0.00 | 0.00 | -0.71 | 0.71 | -0.01 | 0.00 | -0.00 | -0.01 |
| PC-7 | 0.71 | -0.01 | 0.00 | -0.00 | 0.00 | -0.71 | -0.00 | -0.00 |
| PC-8 | 0.00 | 0.00 | -0.00 | 0.00 | 0.51 | -0.00 | 0.65 | -0.57 |

# C  Signal to background analysis

We show here results for signal to background classification. Note that these results are not intended to be used for discovery since we are assuming the same number of signal and back-

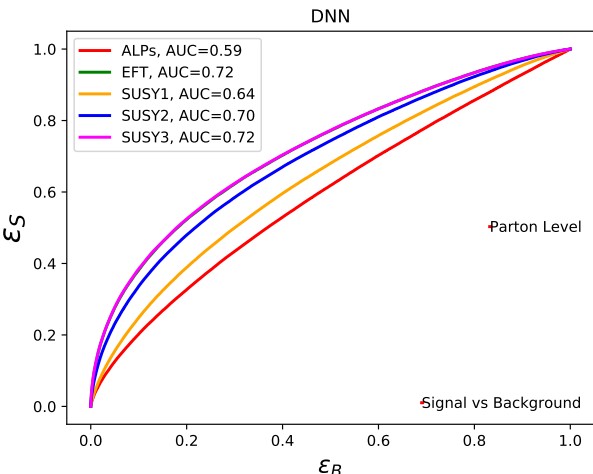

Figure A3: ROC curves using Neural Networks for different signals versus background for the monojet parton-level analysis.

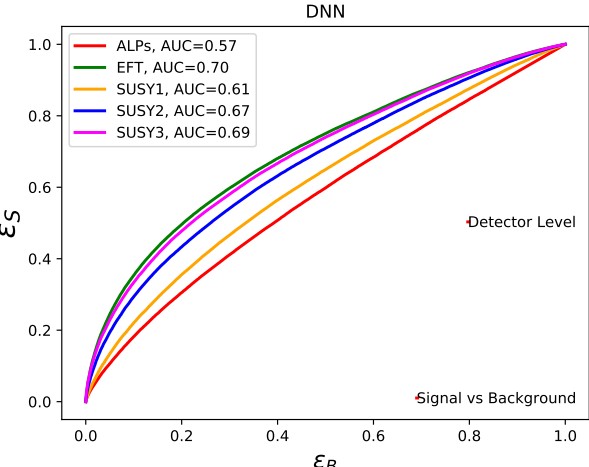

Figure A4: ROC curves using Neural Networks for different signals versus background for the monojet detector-level simulated data.

ground events which would not be the case for real data, but rather to demonstrate the performance of our methods and provide a reference for how well it is possible to distinguish the different signals from background depending on the signal kinematics.

In Fig. A3, we show the ROC curves for various signals at parton-level, whereas the classification accuracy for the detector-level monojet and dijet event generation is shown in Fig A4 and A5 respectively. ALPs are more difficult to pick up from the SM background, a task that becomes simpler and simpler as we increase the mass of the DM particle. SUSY3 and EFT, as expected, exhibit very similar performances, and in Fig. A3 the two lines overlap each other. Nevertheless, the SUSY3 benchmark corresponds to a DM particle of 300 GeV, whereas the EFT corresponds to a DM particle with negligible mass.

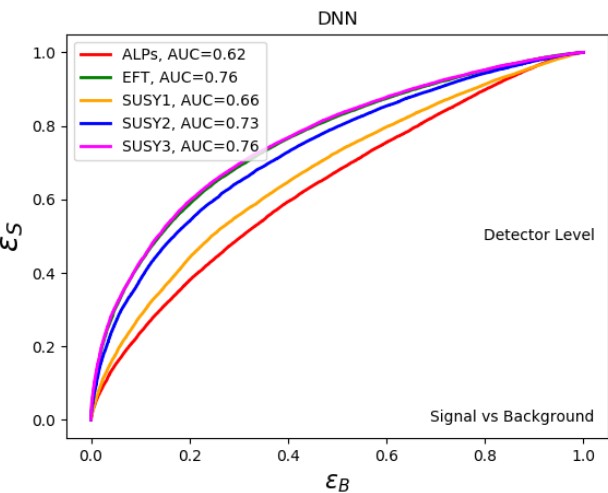

Figure A5: Neural Network dijet detector-level ROC curves for different signals versus background.

# D   Classifier performance at different levels of signal

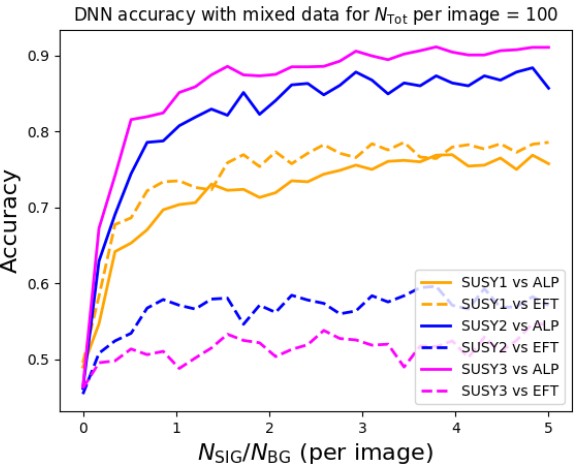

Figure A6: DNN for 2D histogram accuracy for differing number of signal events for trained for 100 epochs.

Here we present how the DNN and CNN for 2D histograms perform for different levels of signal. We run the classifiers as in sections 4.3 and 4.4 but this time we construct the histograms from a mixture of both signal and background events. We set the total number of events per histogram to be fixed and change the ratio of number of signal events $N_{\text{Sig}}$ to background events $N_{\text{BG}}$ and show the model accuracy in Figure A6 for when distinguishing between SUSY WIMP events and ALP and EFT signals for various event ratios. Note that in the limit $N_{\text{Sig}}/N_{\text{BG}} \to \infty$ we reduce to the previous situation where we are comparing only signal with no background. We see that it is harder to distinguish SUSY signals from the EFT than the ALP, especially around the $m_{\tilde{\chi}^0} = 300$ GeV range.

The plot shows us that we require comparable levels of signal to background for good performance, illustrating that this method is not suitable for discovery since discovery would necessitate good performance for $N_{\text{Sig}} \ll N_{\text{BG}}$ (it would be expected that performance increases with total number of events - perhaps with a very large number of events the algorithm would be much more sensitive for lower levels of signal but that is beyond the scope of this paper.

The point of having the mixing between signal and background is to demonstrate how effectively we can disentangle DM candidates beyond the perfect idealised scenario where $N_{\text{Sig}} = N_{\text{BG}}$ - assuming a discovery of signal is made it is good to know benchmarks for performance since we would not be able to fully separate signal from background in a realistic scenario. Knowing these benchmarks is also useful for when constructing unsupervised algorithms where it is useful to know the number of events required for good performance.

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
