# Peer review of "Using Machine Learning to disentangle LHC signatures of Dark Matter candidates"

_SciPost Physics, doi:SciPost Phys. 10, 151 (2021)_

## Round 2 · Referee Report · Anonymous (Referee 1) · 2021-1-30

Report

Thank you for taking into account my feedback. There is one major point which I would like to followup on before I can recommend publication:

Q (me): From an information theory perspective, there is no additional information in r images for classification than there is from a single image. It may be that practically, it is useful to train on multiple events, but since they are independent, formally you should not do better. So either (a) you have found something very interesting about the practical training required in this process or (b) you are not doing a fair comparison when you look at r = 1 versus r
>
1.

A (you): We agree that these two situations should be equivalent in theory. Both contain, in different formats, the same information and in the asymptotic limit of an infinite training set we should get the same result. In practice, though, no algorithm learns all the information in the data, but focuses on getting better at performing a task (like classification) in whatever finite-size dataset is provided. In other words, the algorithm does not learn all correlations in the data. What we showed is that the learning could be improved by bunching the events in images.

When we project N events on a single image we are giving the partial information of the likelihood function which helps the DNN to learn more efficiently. On the other hand, when the NN is trained on images containing one event, learning the probability distribution becomes part of the tasks assigned to the algorithm.

One can compare these two situations in the limit of r=1, where we do indeed get the same accuracy in both cases. To illustrate this point, we have updated Fig. 8 (Fig. 9 in the previous version) to include the lower r values, where one can see the accuracy decreasing when we move towards lower r-values and how for r=1, it approaches to the accuracy reported in the Fig.5. We have also clarified this point in the text.

Followup (me): I am confused what accuracy means in Fig. 8. Is this the accuracy of the model trained with a given r? If you compare r = 1 versus r = 2 do you compare networks that gets to use information from one event with networks that get to use information from two events? This would not be a fair comparison so I must be missing something critical for understanding Sec. IV C.
  • validity: -
  • significance: -
  • originality: -
  • clarity: -
  • formatting: -
  • grammar: -

Author:  Charanjit Kaur Khosa  on 2021-04-14  [id 1361]

(in reply to Report 1 on 2021-01-30)
Category:
answer to question

Yes, r=1 and r=2 means that the networks are trained with images containing 1 and 2 events, respectively. As explained in the text and in the caption of Fig 8, each accuracy line corresponds to a total number of events used to create the images. This comparison is precisely what we want to show: networks trained with  images made up from different values of r=events/image do lead to different accuracies. At about 50 events per image the accuracy plateaus, as one would expect. We have added another sentence in the discussion, hoping to clarify further what we have done.

---

## Round 2 · Referee Report · Anonymous (Referee 2) · 2021-2-1

Report

Dear authors,

thank you very much for adressing all questions in detail and taking the feedback into account. I only would like to follow up on two comments.

~~~~

Q: The phi distribution shows a surprising effect: The fluctuations between the samples of different physics models seem to be smaller than the fluctuations between neighbouring bins.

A: This is just an artefact of the chosen bin size.

Q: I think what you see here are detector effects as modelled by Delphes. If you choose a binning that is 'n * number of calorimeter cells' you should observe a clear periodic pattern. This effect breaks the independence of phi.

~~~~

Q: Fig. 10: According to Fig.9, the accuracy clearly increases for larger values of r. Why did you choose a comparably low value of r=20 for this plot?

A: Yes that is right accuracy increases with r. For the fair comparison, we are considering same amount of data (i.e. 50K events) for most of the ROCs. Since total number of events are fixed choosing higher r leaves us with less number of images to train with.

Q: In this case I think it would make sense to simulate a larger number of signal events in general. This would allow you to really profit from large r values.
  • validity: -
  • significance: -
  • originality: -
  • clarity: -
  • formatting: -
  • grammar: -

Author:  Charanjit Kaur Khosa  on 2021-04-14  [id 1362]

(in reply to Report 2 on 2021-02-01)
Category:
answer to question

'I think what you see here are detector effects as modelled by Delphes' We agree, that is the reason behind this behaviour.

' In this case I think it would make sense to simulate a larger number of signal events in general.' We do agree as well, we have added a comment on this in the text.

---

## Round 2 · Referee Report · Anonymous (Referee 3) · 2021-4-2

Report

Dear Authors,
thank you for addressing my comments. I have one more comment below prior to recommending for publication.

Requested changes

While I understand your point that you are testing an ideal case of a discovery where the signal has already been identified, I don't think this is yet sufficiently clear in the introduction and conclusion. Would it be possible for you to add two sentences, one in each, specifying the two points that you outlined in your reply? You wrote some of this already this quite well in the reply: "...assuming an excess of events has been identified and one is trying to unveil its true origin. We then ask the question Could there be contamination of the SUSY imposters in the vanilla SUSY WIMP events?" In any case I leave the wording to you, but the points I'd like to see would be: - that the signal has already been located and identified, and the irreducible background has been reduced to the point of being able to take shape comparisons in ML tests (otherwise it's confusing to include the background in your signal samples as it's not clear whether you want to remove it or not) - that this is a scenario that will occur if DM is discovered at the LHC or at other experiments that point the way to a specific region for the LHC

Thank you!

  • validity: -
  • significance: -
  • originality: -
  • clarity: -
  • formatting: -
  • grammar: -

Author:  Charanjit Kaur Khosa  on 2021-04-14  [id 1363]

(in reply to Report 3 on 2021-04-02)
Category:
answer to question

We have added these sentences at the beginning of Sec 4, clarifying further what was already written there regarding our aim to characterise the DM signal.

---

## Round 2 · Author Response

We would like to thank the referees for their thorough reading of our paper and their thoughtful comments. We believe the modifications they have suggested will improve the quality of  our paper and its usefulness. We have updated the draft to implement (and to clarify) suggestions from the referees.  The pointwise reply to the referee's comments will be uploaded in the `reply to referee' section.

---

## Round 2 · List of Changes

1. Introduction: we added several new sentences and references, and clarified the text at several points.
  2. Removed figure 2 (in the previous draft).
  3. Updated figure 2, 3 and 7 (in the new draft) to include units.
  4. Section 3: removed the confusing discussion about the 2D images.
  5. Updated figure 8 (this number in the new draft) to include lower values of $r$ upto 1. We also added the corresponding discussion in section 4.
  6. Section 4: we rewrote several sentences and added new text for the clarification.
  7. Added two new figures 14 and 15 in the discussion and the text explaining them.
  8. We compared the background simulation with the latest ATLAS monojet paper and added the text about this comparison in the Appendix.
  9. Other than the above changes there were minor clarifications and changes at many places in the paper.

---

## Round 3 · Referee Report · Anonymous (Referee 1) · 2021-4-28

Report

Thank you for clarifying. I am now quite confused. The r = 2 classifier has different information than the r = 1 classifier and yet the r = 2 classifier is a suboptimal use of the information. If you use the r = 1 classifier, but apply it to 2 events at a time, my guess is that its accuracy will be much better than the r = 2 classifier performance. In fact, as you consider n -> \infty events at a time, the accuracy should converge to 100% as long as the r = 1 classifier is not random. This makes me wonder how you made Fig. 15. You reference [33] which says "We have then derived the signal significance for each signal region from the number of signal events (S), after having imposed the cuts above, and the number of background events (B)". However, in your case, you have a classifier that only operates at the level of 20 events at a time. Can you please clarify? Also, do you have a baseline for Fig. 15? Are these results better than a baseline?

I appreciate that you would like to make minimal changes to the paper at this point and I would support this. Perhaps you could add a couple of sentences to further clarify the above points?

---

## Editorial Decision

published